# Intersectoral Cost of Treating Pulmonary Non-Tuberculosis Mycobacterial Disease (NTM-PD) in Germany—A Change of Perspective in Disease Management

**DOI:** 10.3390/ijerph16203795

**Published:** 2019-10-09

**Authors:** Roland Diel, Pontus Mertsch

**Affiliations:** 1Institute for Epidemiology, University Medical Hospital Schleswig-Holstein, Niemannsweg 11, 24015 Kiel, Germany; 2LungClinc Grosshansdorf, Airway Research Center North (ARCN), Member of the German Center for Lung Research (DZL), 22927 Großhansdorf, Germany; 3Institution for Statutory Accident Insurance and Prevention in the Health and Welfare Services (BGW), 22089 Hamburg, Germany; 4Department of Internal Medicine V, Ludwig-Maximilians-University of Munich, Comprehensive Pneumology Center (CPC-M), Member of the German Center for Lung Research (DZL), 80336 Munich, Germany; Pontus.Mertsch@med.uni-muenchen.de

**Keywords:** cost analysis, Monte Carlo simulation, non-tuberculosis mycobacteria, Mycobacterium avium, integrated care, healthcare management

## Abstract

Background: In line with its increasing prevalence, pulmonary Mycobacterium avium complex (MAC) disease (MAC-PD) gives rise to substantial healthcare costs. However, there is only limited information on the costs of intersectoral reimbursement. Objectives: Inpatient and outpatient costs for diagnosing and treating pulmonary MAC-PD in Germany in accordance with standard international guidelines were calculated and their potential effects on MAC disease management in Germany were determined. Methods: Hospitalization costs were calculated by using the German diagnosis related group (G-DRG) browser, with and without inclusion of the diseases most often associated with M. avium. Separated by drug macrolide susceptibility and severity of MAC-PD, the direct medical costs of suitable therapies in the outpatient setting were determined by Monte-Carlo simulation, including all conceivable options. Results: According to our simulation, the weighted mean cost of outpatient treatment over 14 or 18 months, in either case followed by a post-treatment monitoring over 12 months, amounts to €8675.22 (95% confidence interval [CI] €8616.17 to €8734.27). Of that amount, the revenue for outpatient doctors´ services, dependent on treatment duration, is low, ranging between €894.79 (10.3%) and €979.42 (11.3%), accordingly. Mean drug costs for MAC-PD patients amount to €6130.25 [95% CI €6073.52 to €6186.98], i.e., more than two third (70.7%) of the total outpatient costs. In contrast, the non-surgical reimbursement for a hospital stay of up to 14 days is €3321.64. Hospital reimbursement does not increase in cases of complications (a higher number and/or challenging type of associated diseases), but it is fully paid even in cases that require as few as 2 days of hospitalization. Conclusion: The imbalance between well-rewarded hospital care and the low reimbursement for long-term treatment of MAC-PD outpatients may induce inappropriate disease management. In order to arrive at properly integrated care of MAC-PD patients in Germany, measures such as better incentives for physicians in the outpatient setting and a targeted use of resources in hospitals are required. Reimbursed, periodic case conferences between outpatient physicians and experts in hospitals as well as preventive short-term checks of MAC-PD patients in specialty clinics may promote cross-sector cooperation and improve overall treatment quality. Nationwide pilot studies are required to gain evidence on the effectiveness of the new approach.

## 1. Introduction

Pulmonary disease caused by non-tuberculous mycobacteria (NTM-PD) is a growing, worldwide public health challenge [1], whereby species of Mycobacterium avium complex (MAC), i.e., M. avium, M. intracellulare, and M. chimaera, are the most common pulmonary NTM pathogens in almost all regions of the world [2,3]. 

The annual increase in NTM-PD prevalence in recent years has been shown to be more than 8% in the United States [4], and recently, Ringshausen et al. observed a significant increase of 5.9% per year in overall age-adjusted NTM-PD-associated hospitalizations in Germany [5]. In South Korea, the age-adjusted NTM prevalence is estimated to be 33.3 per 100,000 population in 2016 [6].

NTM-PD-associated morbidity is characterized by creeping destruction of the patient’s lungs, and places significant economic burdens on the health care system. Based on administrative data of the German statutory health insurance community (SHI), the mean direct expenditure per NTM-PD patient in the 39 months following first diagnosis was €39,559.60, nearly four times that for a matched control [7]. During this observational period, a clearly higher mortality rate occurred in patients with NTM-PD (22.4%) as compared to patients of the control group (6%) (*p* < 0.001). 

Hospital costs are three times higher in the NTM-PD group than in the matched non-NTM-PD group. However, most patients (52%) are initially diagnosed and treated as outpatients (unpublished original data for [7]) although in Germany there is generally no waiting time for access to pulmonological hospital departments or lung clinics. 

Of note, during the second year of treatment, 31.1% of the surviving NTM patients had to be either hospitalized for the first time or re-hospitalized [7], suggesting that the choice of medication and/or treatment adherence in the NTM-PD group may have been, at least in part, ineffective. In effect, hospitalization proves to be the main cost driver in resolving NTM-PD cases.

According to the available SHI data, only 54.4% of patients began treatment within three months of a new NTM-PD diagnosis, and median time between (discharge) diagnosis and therapy initiation was 32 days (interquartile range [IQR] 260 days), revealing some uncertainty about which basic regimens should be chosen [7]. This is of high importance, as a recently published study analyzing data from a large US managed care provider claims [8]. There, guideline base treatment (GBT) was associated with a significantly lower all-cause hospitalization risk versus non-GBT (odds ratio 0.53) and adjusted total health-care expenditure in year 2 with GBT was clearly lower than that with non-GBT, the difference being $7933.

The high economic burden of pulmonary NTM disease in Germany raises the question as to how better disease management may contribute to improved clinical outcomes. An essential structural feature of the German health care system is its strict separation into two different sectors, the outpatient and the hospital sector. Although both healthcare provider types are paid by the SHI, reimbursement to each sectors follows the sector’s own system and accounting procedures, namely the Statutory Health Insurance Scheme (Einheitlicher Bewertungsmaßstab, EBM) [9] for ambulatory services and the German diagnosis related complex system (G-DRG) for hospital stays, updated each year by the InEK (Institute for the Hospital Remuneration System [10]). A special form applicable to cases of MAC-PD, the so-called ambulatory specialized care (“ambulante spezialärztliche Versorgung, ASV”), is defined in paragraph 116 b SGB V [Volume V of the Social Insurance Code (Fünftes Buch Sozialgesetzbuch—SGB V)] and was implemented in 2012 for continuing tuberculosis (TB) or NTM-PD treatment. It may be used in both sectors, but is applicable only under the EBM system [11].

To date, the only data available for the costs in Germany of treating NTM patients, Diel´s burden of illness study, [7] is confounded at least in part by the inclusion there of patients with concomitant diseases in the NTM group as well as in the matched non-NTM group. A systematic analysis of the costs occurring exclusively for diagnosing, monitoring, and treating of NTM-PD in the two German sectors, including the inherent problems of each, has not, to date, been published. Our analysis also aims to clarify options for overcoming structural weaknesses in the treatment of NTM-PD, based on the example of Mycobacterium avium complex, the most commonly isolated pathogen in NTM-PD worldwide. 

## 2. Material and Methods

This analysis considers the charging of direct medical costs under the current status quo for diagnosis and therapy of pulmonary MAC disease as accepted and borne by the SHI. 

Our calculations of the direct costs of MAC-PD outpatients are therefore based on the drug costs occurring under different treatment combinations and on the rates established by the Statutory Health Insurance Scheme (EBM) [9] for outpatient services. Costs of inpatients were assessed using data provided by the InEK for the G-DRG system [10]. 

According to international recommendations, four different drug combinations for a standard treatment of macrolid-susceptible MAC-PD cases, two for macrolid-resistant MAC-PD and four for severe MAC-PD, have to be taken into consideration (for details see Appendix A). The drug costs of each treatment option were determined and the assumed frequency of these options (probability weight) was applied. To calculate the mean value and the 95% confidence interval (CI) of the drug costs of all treatment options, a “random walk” through the 10 different options was made by Monte Carlo simulation [12], with 10,000 simulation runs performed. 

Subsequently, the drug costs of the respective treatment regimens were each added to the separately calculated costs of treating and monitoring outpatients, either for 26 or 30 months (see below for details), arriving at 20 different options. Again, the probability weight of each option was applied, whereby—due to uncertainty on how often a shorter or longer treatment is required in a pulmonary MAC patient—the two treatment durations were given the same probability (see Table 2). To calculate the mean value and the 95% confidence interval (CI) of the total outpatient costs, a random walk was made through the 20 different options by a second Monte Carlo simulation, again with 10,000 simulation runs performed. 

### 2.1. Basic Assumptions for Calculating Outpatients Drug Costs

As a first step in our modeling, we assume that general practitioners refer afflicted patients either directly to a hospital, or to a lung specialist in a private practice. Accordingly, the patient will undergo diagnostic procedures and therapy, either as a primary inpatient or as a primary outpatient throughout the treatment and port-treatment monitoring period. Drug costs for outpatients are determined by selecting the least expensive alternatives from the pharmacy retail price catalogue (Apothekenverkaufspreis, AVP), which is part of the Red List (Rote Liste^®^) 2019, Germany’s register of pharmaceutical drugs [13]. Discount agreements for drug costs, an option open to single health insurance organizations under the German system, are not considered in our calculations. The drugs and the abbreviations used in the model are listed in Table 1. Daily therapy costs were calculated by dividing pack quantity by the drug’s recommended daily intake as stated in the dosing instructions.

Unfortunately, patterns of resistance in MAC-PD have not yet been systematically studied, and data in the literature are sparse. Generally, a selection of three of six drugs taken daily is recommended for the entire 14 or 18 months of treatment. The choices are a macrolide (clarithromycin (CLAM) or azithromycin (AZM)), ethambutol (E), and rifampicin (R) or rifabutin (RBT) in macrolide-susceptible MAC-PD patients, and in macrolide-resistant cases, clofazimin (CLO) or a fluorquinolone, e.g., moxifloxacin (MOX), instead of a macrolide. Amikacin (AMK) should be administered intravenously for at least eight weeks in macrolide-resistant and/or severe MAC-PD cases (for details see Appendix A). As sputum conversions generally occur only after 2–6 months of treatment, and since 12 months of further treatment should follow conversion, the typical MAC-PD patient will receive treatment for 14 to 18 months. 

As the actual composition of drug regimens administered to MAC-PD patients in Germany is not known, weighting of possible treatment schedules was required in our modeling. From the literature, we found that Medical records of 634 human immuno-deficiency virus [HIV]-negative patients with pulmonary MAC disease in Japan showed fibrocavitary disease in 16.6% of patients [14]. Griffith et al. [15] reported a similar rate of macrolide resistance (15.92%) in 565 patients with MAC lung disease. Thus, we assumed that generally 84% of MAC patients could be treated without AMK. For the remaining 14%, the probabilities of the possible regimens were considered to be equally distributed (see Table 2). 

### 2.2. Assumed Procedures in the Diagnosis and Monitoring of MAC-PD

1. Microbiology:

Initially, to verify that the strain represent disease rather than colonization, three separate expectorated sputum samples were collected on separate days for microscopy and sent for culturing. The procedure was repeated under treatment every four weeks until negative results were obtained, i.e., at least five times for a presumed conversion within eight weeks or nine times for a presumed conversion within 6 months. The first positive culture in the initial phase is subjected to resistance testing.

NTM-PD patients must be strictly monitored: following culture conversion, sputum smear microscopy and culture examination are performed at least every 3 months (BTS) throughout the 12-month continued drug therapy and subsequently every 3 months for 12 months, i.e., eight times. In total, the number of microscopies and cultures before, during and after treatment, amounts to 13 or—if culture conversion cannot be achieved within 6 months—to 17 repetitions. 

2. Blood work:

A broad blood profile was necessary before therapy started. Kidney retention values (creatinine, urea) and liver values (glutamic oxaloacetic transaminase [GOT], glutamate pyruvate transaminase [GPT], bilirubin, gamma-glutamyltransferase [GGT]) are key. GGT is required as an alcohol abuse parameter and also allows as an indicator for Rifampin-induced cholangitis. It is also advisable to establish a serological profile for hepatitis (hepatitis B virus surface antigen [HBs-Ag], antibodies to hepatitis B core [Anti-HBc]) and HIV. Liver values must be controlled 2–4 weeks after starting therapy, and every 4 weeks thereafter. The renal retention parameters are checked every month through to the end of the therapy, together with the liver values. 

Serum electrolytes and CRP are checked every 2 weeks for 2 months, then monthly in all patients. If intravenous AMK has been prescribed as an additional drug, the serum level of AMK should be controlled at the end of the first week, and again 2 weeks and 6 weeks after starting the therapy (given normal kidney function). 

3.Ophthalmic examination: under EMB, before therapy starts and usually every 4 weeks, i.e., 14 to 18 times in total.If AMK has been prescribed, audiometry was performed in the initial phase and then every 4 weeks during therapy, altogether three in total.4.X-rays are recommended in the initial phase, after 4 weeks (to control if the NTM is reacting to the therapy) and after 8 weeks (end of the initial therapy: success assessment). After 8 weeks, X-ray checkups in the 4th and 6th month are sufficient, followed by check-ups every 3 months and 6, 12, and 24 months after the end of treatment, amounting to 11 in total. The number of X-rays increases to 15 if culture conversion cannot be achieved with 6 months of treatment.5.Computer tomography will be used as a diagnostic supplement prior to therapy and then every 6 months, i.e., three times in total.6.ECG monitoring has to be performed when using drugs that may prolong corrected QT interval [QTc](CLAM or AZM). ECG is recommended at 0, 2, 12, and 24 weeks.

## 3. Results

### 3.1. Outpatient Costs

The costs of all reasonable treatment regimens under consideration, separated by drug costs for a treatment of 14 or 18 months, and the remaining costs are depicted in detail in Table 2. According to the results of the first order Monte Carlo simulation that takes account of treatment variability, the weighted mean drug costs were €6130.25 [95% CI €6073.52 to €6186.98]. The remaining costs of diagnosing, treating, and monitoring pulmonary MAC disease (under the standard course of therapy) are €2162.47 for a 14-month course of treatment and €2895.75 for 18 months. The addition of intravenous AMK in severe or macrolide-resistant cases, increases the costs to €2263.3 for a 14 month and €2996.58 for a 18 month course of treatment (see Table 3). The weighted mean of outpatient treatment cost over 14 or 18 months, followed by a post-treatment monitoring over 12 months, as calculated by the second Monte Carlo simulation, is €8675.22 (95% CI €8616.17 to €8734.27).

Of that amount, the revenue for outpatient doctors´ services themselves only plays a minor role; in Germany, the outpatient doctors are paid per quarter. Thus, for the shorter treatment duration the revenue refers to nine quarters, or 27 months, and for the longer treatment duration to 10 quarters, or 30 months. According to the figures presented in Table 4, the revenue for the GP is €517.14 for the shorter treatment of 14 months plus the 12-month monitoring period and €574.60 for the longer treatment duration of 18 months. The pulmonologists receive €377.65 and €404.82, respectively. Dividing these revenues by 26 or 30 months the (rounded) revenue per month for the GP is on average €19.89 (€517.14/26) and €14.53 (€377.65/26) for the pulmonologist or €19.15 (€574.6/30) and €13.49 (€404.82/30). That means that—dependent on treatment duration—the revenue for outpatient doctors (adding the revenues for the GP and the pulmonologist together) per month is on average €34.42 or €32.65.

Summing up the revenue for the GP and the pulmonologist to one figure, according to Table 4, the percentage share of the costs for outpatient doctors´ services of the weighted mean of total outpatient costs is 10.3% (€894.79/€8675.22) (10.3%) for a total of 26 months or 11.3% (€979.42/€8675.22), for a total of 30 months. 

### 3.2. Specialized Ambulatory Care (ASV)

Since 2012, ambulatory specialized care for TB and NTM can be conducted in Germany under strict conditions in accordance with section 116b SGB V in both the outpatient setting and hospitals. However, contracted physicians and hospitals intending to implement such an ASV must provide their services at the same (low) rates as with purely outpatient remuneration (EBM). 

Furthermore, the prerequisites for maintaining an ASV are very strict; per year, at least 20 patients with TB or NTM have to be treated in an interdisciplinary team with a pulmonologist as director and under settled doctor’s conditions. Beyond the outpatient-based remuneration, full-time equivalents for doctors or nursing in the hospital that establishes such a service will not be paid. Thus, a hospital can cover the actual costs of such ambulatory care only with a pre-existing high organizational overhead. On the other hand, the number of new TB or NTM cases a single outpatient doctor will see in a given year is generally well below the 20 or more required for an ASV. 

Accordingly, for all participants, whether hospital or outpatient doctors, the procedure of furnishing the continually requested proof that the conditions of participation can be fulfilled is burdensome. Therefore, it is no surprise that the willingness to engage in ambulatory specialized care is very low; to date, only 34 ASV for treating mycobacterial diseases currently exist in Germany, of those 27 in hospitals [9]. 

### 3.3. Hospital Reimbursement

As far as in-patients are concerned, since January 1st, 2004, hospital costs are based on the uniform German G-DRG system, which allocates each case to a diagnosis-related group. Under that system, reimbursement of hospital services is no longer based on fixed daily rates for the period of stay, but focuses on the type and severity of the diseases and the coding of the medical operations and procedures according to the German Procedure Classification (Operation and Procedure Code, OPS), e.g., diagnostic bronchoscopy (OPS 1-690.0) or thoracic computer tomography (OPS 3-222), performed for the respective NTM patient during the hospital stay. Diagnosis of the respective diseases of the inpatients, with the associated procedures performed, must be coded by the responsible physicians using the respective German (G)-ICD 10 list and entered in an officially approved grouper (GetDRG [16]). A national base rate for the hospitals that is updated annually by the InEK is then multiplied by the specific cost weight of a disease that will be calculated according to an internal algorithm of the IneK, resulting in the effective reimbursement for the hospital. Usually, a hospital stay of NTM-PD patients in Germany is 14 days [10], of which the admission and discharge day are each credited only as half and pulmonary NTM disease (ICD-10 A31.x, infections by other mycobacteria) will be grouped into the DRG E76C (TB without severe complications, hospital stay up to 14 days).

Of note, with the exception of cystic fibrosis (CF), inserting all reasonable combinations of NTM as main diagnosis in the grouper together with well-known associated or underlying diseases (e.g., all degrees of chronic obstructive pulmonary disease [COPD] (J44.0x) [17], bronchiectasis (J47), asthma (J45.1), sequela of prior pulmonary TB (B.90.9), Sjögren syndrome (M35.0), gastro-esophageal reflux disease (K21.0) will not change the DRG and accordingly not the cost weight.

In 2019, the national base rate is €3544.97 [18]. By multiplying that amount with a cost weight of 0.937 for G-DRG E76C, the mean effective reimbursement (without considering administrative surcharges such as for being a training center) for treating a non-CF NTM over 14 days is €3321.64. That amount will not change with increasing or decreasing age, gender, or the number of performed OPS. Reimbursement only increases in the rare cases when surgery due to NTM-PD is required in differential diagnosis to neoplasms of obscure origins or in case of destroyed lung lobes, due to specific surgical OPS and postoperative ventilation.

Of note, coding A.31x (NTM as a secondary diagnosis (ND) does not increase the relative weight in all selected scenarios.

To calculate the average daily reimbursement per hospital bed the reimbursement of 3321.64 has to be divided by mean length of stay for E76C. Using the respective values as described in [16] results in (€3544.97 × 0.937)/6.4 = €519.0.

## 4. Discussion

To our knowledge, our study analyzes for the first time the reimbursement structure for the treatment of pulmonary NTM disease in a high-income country. In particular, we have compared the different modes used in the outpatient and inpatient setting and their economic effects. As our cost calculations demonstrate, done separately for each of the two sectors of healthcare services in Germany, guideline-based diagnosing and treatment of MAC-PD in the outpatient setting is low-paying, with the bulk of reimbursement (70.7%) going to medication. Reimbursement for physicians´ services to a MAC-PD does not exceed 11.3% of the total, which is equal to €32.65 per month. By comparison, the reimbursement for a hospital stay appears attractive; the total amount for a non-surgical NTM-induced hospital stay is “capped” by the internal algorithm of the G-DRG grouper to €3321.64, irrespective of the number and type of NTM-associated diseases the respective patient is suffering from, but covers 38.3% per cent of the whole outpatient reimbursement, and that in 14 days rather than in 30 months. 

Our findings may have important implications for the outcome of NTM treatment; namely, that the discrepancy between presumably well-rewarded hospital care and the remarkably low reimbursement for long-term outpatient treatment probably hampers co-operation between physicians in the two sectoral “silos” and may contribute to the generally poor outcomes of the disease in Germany. Diel´s publication [2] revealed an increase in mid-treatment hospital stays for German patients with NTM-PD, so that on average every NTM patient is hospitalized once a year. However, it remains unclear whether the responsibility for the rapid progression of disease that leads to hospitalization should be primarily attributed to delayed initiation of treatment with inappropriate drugs, treatment interruptions due to adverse events, or negligence in adherence on the part of the patient´s. Of note, in Germany, all costs for diagnosing and treating NTM-PD—with exception of a fee of €10 per hospital day for no longer than 28 hospital days per year—are free of charge to patients. Expenses for travelling to outpatient doctors as well as to a hospital are covered by the SHI in the same vein [19], so that additional costs to be borne by the patient (out of pocket expenses) are not expected to play a significant role with respect to a preference for either inpatient or outpatient treatment. New PCR tests may be useful for the rapid diagnosis of mycobacterial infection and differentiation of MTB from NTM in microbiologically positive NTN patients, as they support immediate initiation of treatment, but their use is still limited by a lack of awareness on the part of both inpatient and outpatient doctors of NTM-PD as a probable cause of pulmonary disease.

The German outcome data suggest a clear need for integrated care, with consulting and coordination between pulmonological expert centers, i.e., specialized lung clinics and outpatient doctors and beginning with the diagnosis of NTM-PD. However, how should such integrated care that aims to establish more patient-oriented care and cross-sectoral communication look in reality? A useful example exists in the pulmonology community in Europe: a most recently published systematic review of integrated disease management showed that interventions in COPD patients from 11 industrial countries (except of Germany), involving at least two different categories of healthcare providers, reduced respiratory-related hospital admissions on the short term by 32% and on the long-term (>12 months) by 41% [20]. Unfortunately, this example has not yet been applied to the apparent need for integrated care on NTM-PD.

Another example is a Germany-specific “lighthouse project” established in 2005 according to § 140 SGB V, which addresses the full spectrum of morbidities and health issues for a population: “Gesundes Kinzigtal” (Healthy Kinzigtal) [21] is named for the largely residential area it covers. The Kinzigtal region, located in the Black Forest in Southwest Germany, is home to about 70,000 people. There, a joint management company of regional physicians and the participating SHI coordinates patient care between health care providers and distributes eventual cost savings achieved between the contractual partners. The Healthy Kinzigtal approach could effectively be applied to the management of NTM disease. 

The two examples, however, bring to light the main limitation of our findings: they do not provide any proof that integrated care will also result in improved outcome in pulmonary NTM disease. To gain evidence of the advantages closer alignment of the inpatient and outpatient sector may provide, future research is urgently required. An exemplary implementation and trial of optimized care for pulmonary NTM patient should be started in the near future, supported by several major SHI, allowing nationwide participation by pulmonological expert centers for NTM treatment and their referring outpatient physicians.

To achieve integrated care for NTM-PD patients, periodic case conferences between specialized hospital colleagues and outpatient doctors, coupled with voluntary 2-day hospital check-ups twice a year, could form the core of a new model. With the objectives of maintaining the quality of life of patients, re-assessing the course of disease and to preventing pulmonary exacerbations, these two days should be sufficient to complete all scheduled investigations, such as thoracic computed tomography (TCT), bronchoscopy, and sampling for microbiological investigations. Given that the amount of €519 per day—as calculated above—would provide adequate reimbursement to the hospital, the funds required for such a program would be less than the presently expected cost of punctual hospitalization for the management of pulmonary exacerbations that most NTM-PD patients experience. Given the COPD experience, where reductions of 32% to 41% in hospitalization due to exacerbations have been reported, the savings should be enough to provide incentives for settled physicians and finance the intersectoral exchange, inclusive of additional administrative expenses. For conducting such pilot projects, an accompanying evaluation should be established to enable a pre-post comparison of routine SHI data.

In conclusion, integrated NTM-PD care in Germany is urgently required to improve overall treatment quality and to avoid unnecessary hospitalizations due to exacerbation. An economic trade-off between the inpatient and outpatient sector is one of the prerequisites for achieving that target. To gain evidence of that that new approach, further research in terms of nationwide pilot studies is necessary.

## Figures and Tables

**Table 1 ijerph-16-03795-t001:** List of medical drugs for the treatment of pulmonary disease caused by non-tuberculous mycobacteria (MAC-PD) available in Germany.

Name	Abbreviation	Form/Price	Daily Dose
Amikacin FRESENIUS	AMK	10 Infusion bottles - Fl. 500 mg/100 mL: €357.0	2×1: €71.40
Clarithromycin-ratiopharm 500 mg	CLAM	20 film-coated tablets: €19.52	2×1: €1.95
Zithromax 250 mg	AZM	6 film-coated tablets: €20.77	1× 1: €3.46
Clofazimin (LAMPREN) 100 mg	CLO	100 tablets: €199.92	1×1: €2.0
Ethambutol (EMB-Fatol) 400 mg	E	100 film-coated tablets: €41.75	3×1: €1.25
Moxifloxacin HEXAL 400 mg	MOX	10 film-coated tablets: €39.73	1×1: €3.97
Eremfat 600 mg	R	100 film-coated tablets: 271.46	1×1: €2.71
Rifabutin MYCOBUTIN PFIZER 150 mg	RBT	90 capsules: €489.19	2×1: €10.87

**Table 2 ijerph-16-03795-t002:** Total costs of diagnosing, monitoring, and treating pulmonary MAC disease in German outpatients.

Medication (Possible Drug Combinations)	Drug Costs/Day in €	Drug Costs 14 Months in €	Drug Costs 18 Months in €	Costs of Diagnostics/Monitoring 14 Months in €	Costs of Diagnostics/ Monitoring 18 Months in €	Total Outpatient Cost 14 Months in €	Total Outpatient Cost 18 Months in €
Macrolid-susceptible MAC-PD							
R-E-CLAM	5.91	2482.2	3191.4	2162.47	2895.75	4644.67 (0.84/8)	6087.15 (0.84/8)
RBT-E-CLAM	14.08	5913.6	7603.2	2162.47	2895.75	8076.07 (0.84/8)	10,498.95 (0.84/8)
R-E-AZM	7.42	3116.4	4006.8	2162.47	2895.75	5287.87 (0.84/8)	6902.55 (0.84/8)
RBT-E-AZM	19.54	8206.8	10,551.6	2162.47	2895.75	10,369.27 (0.84/8)	13,447.35 (0.84/8)
Macrolid-resistant MAC-PD							
R-E-CLO-AMK (AMK 60 days)	5.96 (4284)	6787.2	7502.4	2263.3	2996.58	9050.5 (0.16/12)	10,498.98 (0.16/12)
R-E-MOX-AMK (AMK 60 days)	7.93 (4284)	7614.6	8556.2	2263.3	2996.58	9877.9 (0.16/12)	11,552.78 (0.16/12)
Severe NTM							
R-E-CLAM-AMX (AMK 60 days)	5,91 (4284)	6766.2	7475.4	2263.3	2996.58	9029.5 (0.16/12)	10,471.98 (0.16/12)
RBT-E-CLAM-AMX (AMK 60 days)	14.08 (4284)	10,197.6	11,887.2	2263.3	2996.58	12,460.9 (0.16/12)	14,883.78 (0.16/12)
R-E-ATZ-AMX (AMK 60 days)	7.42 (4284)	7400.4	8290.8	2263.3	2996.58	9663.7 (0.16/12)	11,287.38 (0.16/12)
RBT-E-ATM-AMX (AMK 60 days)	19.54 (4284)	12,490.8	12,840.2	2263.3	2996.58	14,754.1 (0.16/12)	15,806.78 (0.16/12)

Legend: AMK: Amikacin; E: Ethambutol; CLAM: Clarithromycin; AZM: Azithromycin; CLO: Clofazimin; MOX: Moxifloxacin; R: Rifampicin.

**Table 3 ijerph-16-03795-t003:** Weighted mean of outpatient costs *.

Statistic	Value
Mean	€8675.22
Standard deviation	± €3012.67
Minimum	€4644.67
Median	€8076.07
95% Confidence Interval	€8616.17 to €8734.27
Maximum	€15,806.78
Sum (n × Mean)	€86,752.202.82
Size (n)	10,000
Variance	€9076,177.69
Variance/Size	€907.62
SQRT[Variance/Size]	€30.13

* Cost weight calculated by using Monte Carlo simulation.

**Table 4 ijerph-16-03795-t004:** German statutory health insurance community (SHI) costs of diagnosing and monitoring pulmonary NTM in the outpatient setting.

Medical Services	No. of Points	Individual Payment (€)	Frequency	Payment (€)
**General practitioner**				
Flat rate coverage (irrespective of number of visits by patients) per quarter; (EBM Gebührenordnungsposition (GOP) 03000	157	16.99	9 [quarters] (10)	152.91 (169.9)
GOP 03220, additional fee on GOP 03000 (due to NTM as chronic illness, “Chronikerzuschlag”)	130	14.07	9 (10)	126.63 (140.7)
GOP 03222, additional fee on GOP 03220	10	1.08	9 (10)	9.72 (10.8)
Detailed conversation (GOP04230)	90	9.74	9 (10)	87.66 (97.40)
Retainer fee (general practitioner), once per quarter (GOP 03040)	144	15.58	9 (10)	140.22 (155.80)
**Pneumological diagnostics (NTM)**				
Pneumological consultation; 60 years of age and above (GOP 13642)	210	22.73	9 (10)	204.57 (227.30)
Surcharge (pneumologist), once per quarter (GOP 13644)	41	4.44	9 (10)	39.96 (44.40)
Bronchoscopy (GOP 13662)	988	106.93	1	106.93
Pneumologist BAL (GOP 13663), additional fee on GOP 13662	242	26.19	1	26.19
**Methods**				
ECG (EBM 27320)	Cannot be charged separately			
Audiometry (OPS 09320), ENT code can also be applied (before starting therapy with macrolides)	147	15.91	3 (initially and then once per month)	47.73
X-ray (EBM 34241; a consultation cannot be charged)	152	16.45	11 (15)	180.95 (246.75)
Ophthalmologic consultation (EBM 06212)	150	16.23	14 [once per month] (18)	227.22 (292.14)
Computerized tomography (EBM 34330)	660	71.43	3 [initially and once per half year] (4)	214.29 (285.72)
Surcharge contrast agent (EBM 34345)	228	24.68	3 (4)	74.04 (98.72)
**Microbiology**				
Microscopy test for mycobacteria (EBM 32176)	-	5.20	13 (17)	67.6 (88.4)
NAAT (EBM 32825)	-	61.40	1	61.40
Culture test for mycobacteria (EBM 32747) per material	-	34.90	13 (17)	453.7 (593.3)
Differentiation of mycobacteria (EBM 32764) if positive	-	28.40	1	28.40
Resistance definition (EBM 32770) per mycobacteria type (repeating of susceptibility testing only required if patient recultures MAC after culture conversion)	-	39.50	1	39.50
**Laboratory investigation**				
Anti-HBc (EBM 32614)	-	5.90	1	5.90
HBs-Ag (EBM 32781)	-	5.50	1	5.50
Chlorid (EBM 32084)	-	0.25	16 [every 2 weeks for 2 months, then once per month] (20)	4 (5)
Natrium (EBM 32083)	-	0.25	16 (20)	4 (5)
Potassium (EBM 32081)	-	0.25	16 (20)	4 (5)
Calcium (EBM 32082)	-	0.25	16 (20)	4 (5)
Creatinine (Jaffe method) (EBM 32066)	-	0.25	16 (20)	4 (5)
Urea (EBM 32065)	-	0.25	16 (20)	4 (5)
Blood count (EBM 32122)	-	1.10	16 (20)	17.6 (22.20)
Bilirubin total (EBM 32058)	-	0.25	16 (20)	4 (5)
Gamma-glutamyl transferase (EBM 32071)	-	0.25	16 (20)	4 (5)
Glutamate-oxaloacetate transaminase (EBM 32069)	-	0.25	16 (20)	4 (5)
Glutamate-pyruvate transaminase (EBM 32070)	-	0.25	16 (20)	4 (5)
CRP (EBM 32460)	-	4.90	16 (20)	78.4 (98)
Amikacin serum levels (EBM 32341)	-	17.70	3	53.10

Anti-HBc = Hepatitis B core antibody; BAL = bronchoalveolar lavage; ENT = Ear Nose Throat; HBs-Ag = surface antigen of the hepatitis B virus; NAAT = Nucleic Acid Amplification Test.

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
