# Peer review of "Intersectoral Cost of Treating Pulmonary Non-Tuberculosis Mycobacterial Disease (NTM-PD) in Germany—A Change of Perspective in Disease Management"

_ijerph, 2019, doi:10.3390/ijerph16203795_

Round 1

Reviewer 1 Report

The authors highlight the important but often neglected topic of financing of chronic disease management as it relates to NTM-PD in Germany.

The background detail is relevant and important and the findings of this modelling study are well presented. However, I am unsure that the data presented in the results is sufficient to justify the conclusions drawn by the authors.

It seems as though there are more (important) questions related to diagnostic (and treatment) delay as well as post discharge outpatient follow up that are generated by this research. Some of the diagnostic delay (I suspect), relates to the time necessary to culture MAC in the laboratory. The use of PCR based technology could reduce this. How often is this used for MAC diagnosis in Germany?

Another potential delay is the time from positive (MAC) culture til treatment is started. The authors state that most MAC treatment is initiated as an outpatient (unpublished data). Are there typically long waiting times to access speciality outpatient clinics in Germany? This would be important to understand.

Whilst the authors give (extensive) detail on the costs associated with outpatient and inpatient management I was not clear how much of this cost is born by the patient (out of pocket expense). Is this different for inpatient and outpatient care? This may bias the patient (or health care provider) to offer inpatient or outpatient treatment.

The authors argue for integration of NTM patient care and favour an outpatient approach (much like the current WHO TB recommendations), however, is their available evidence to support the efficacy of this approach?

Author Response

It seems as though there are more (important) questions related to diagnostic (and treatment) delay as well as post discharge outpatient follow up that are generated by this research. Some of the diagnostic delay (I suspect), relates to the time necessary to culture MAC in the laboratory. The use of PCR based technology could reduce this. How often is this used for MAC diagnosis in Germany?

Reply: Thank you for this important comment. Indeed, in microbiological positive specimen and positive cultures PCR based determination of NTM species could be accelerated. There is no survey or study available how often laboratories in Germany use such tests, although in our own experience new PCR tests have already been routinely implemented in metropoles. In our view, as least as important as the technical options to speed up the diagnosis of NTM is the awareness of both, inpatient and outpatient doctors, that eventually a pulmonary NTM disease might be present.  We have now added the following sentence in the Discussion section addressing both points: “New PCR tests may be useful for the rapid diagnosis of mycobacterial infection and differentiation of MTB from NTM in microbiologically positive NTN patients, as they support immediate initiation of treatment, but their use is still limited by a lack of awareness on the part of both inpatient and outpatient doctors of NTM-PD as a probable cause of pulmonary disease.”

Another potential delay is the time from positive (MAC) culture til treatment is started. The authors state that most MAC treatment is initiated as an outpatient (unpublished data). Are there typically long waiting times to access speciality outpatient clinics in Germany? This would be important to understand.

Reply: Thank you for thus important question. Indeed, there is generally no waiting time for access to German pulmonological departments. We have highlighted this fact in the Introduction section.

Whilst the authors give (extensive) detail on the costs associated with outpatient and inpatient management I was not clear how much of this cost is born by the patient (out of pocket expense). Is this different for inpatient and outpatient care? This may bias the patient (or health care provider) to offer inpatient or outpatient treatment.

Reply: Thank you, we have now added a sentence in the Introduction section pointing out that all diagnostic and treatment costs are paid by the statutory health insurances (SHI) and that there is also no difference with respect to reimbursement of transport expenses. Thus, out of the pocket costs hardly occur in Germany for treating and monitoring NTM-PD.

The authors argue for integration of NTM patient care and favour an outpatient approach (much like the current WHO TB recommendations), however, is their available evidence to support the efficacy of this approach?

Reply: Thank you for that important comment. In fact, we do not favor one particular approach but recommend to more closely align both sectors closer linkage. We have now added some new passages on the need for achieving more evidence on the topic in the Abstract and the Discussion section and pointed out in the Discussion section that the lack of evidence for our proposals are a limitation of the study.

Reviewer 2 Report

1. A short background statement should be added in the abstract.

2. The number and date of the ethical committee approval should be mentioned in the method section.

3. The statistical analysis section is unclear and should be improved and better described.

4. The discussion section should be re-written and as follows:

The finding(s) of the current study and comparison with previous published (similar) studies
The implication of the findings
The strengths and limitations of the study
The new direction of the future research

Author Response

A short background statement should be added in the abstract.

Reply: Of course, thank you! In fact, the contents of the actual Objective section represents the background of the study. Therefore, we have partially assigned these contents to the “Background” section and inserted a new sentence addressing the “Objectives”:

“Inpatient and outpatient costs for diagnosing and treating pulmonary MAC-PD in Germany in accordance with standard international guidelines shall be calculated and its potential effects on MAC disease management in Germany be determined.”

The number and date of the ethical committee approval should be mentioned in the method section.

Reply: Thank you, we have now inserted the following mini-section before the “Methods” section:

“Ethical considerations

Ethical approval was not necessary as only publicly available secondary data were used.”

The statistical analysis section is unclear and should be improved and better described.

Reply: The statistical section has now been completely re-written regarding the Monte Carlo simulation for purpose of better understanding. For further details we also refer to the Supplementary Material section.

The discussion section should be re-written and as follows:

The finding(s) of the current study and comparison with previous published (similar) studies
The implication of the findings
The strengths and limitations of the study
The new direction of the future research

Reply: The discussion section has now been re-written according to your suggestions.

Reviewer 3 Report

The manuscript entitled „Intersectoral cost of treating pulmonary NTM disease in Germany – a change of perspective in disease management” which has been submitted by Diel and Mersch is an important contribution to the current discussion on the management of increasing pulmonary NTM. By separately calculating the costs of the inpatient-outpatient sector Diel and Mersch show a clear discrepancy that might enhance the generally poor outcome of that disease. The methodology appears to be sound and the conclusions with respect to the suggested implementation of integrated care in Germany to improvsece the collaboration between the two sectors are convincing.

I have only a few minor comments:
Although the title will be longer, I would recommend to write out "NTM". The title would then be: Intersectoral cost of treating pulmonary non-tuberculous mycobacteria disease (NTM-PD) in Germany - a change of perspective in disease management

General comment: Once the abbreviations of the drugs have been provided the authors should use these abbreviations in the text continuously and not jump from abbreviations to a drug's full name and vice versa  

Introduction, third para: One of the characteristics of pulmonary NTM disease is its high mortality. The authors do not mention any word on this fact and should therefore insert a sentence in order to complete the presentation of the high burden of NTM disease also in high income countries.

Introduction, fourth para: Do the “unpublished data” refer to the dataset of reference no. 7? If so, please clarify!

Basic assumptions for calculating drug costs of outpatients, first para: The authors should mention that possible discounts for the drugs of interest in favor of the statutory public health insurances are not included into the calculations.

Hospital management, second last para: Firstly, the authors should provide the full name of “OPS” not only the abbreviation. Secondly, the authors should give some examples of the mostly relevant OPS and provide the numbers of their associated numbers, e.g. for bronchoscopy, thoracic CT etc.

Discussion, first para: The author should provide a short explanation how the figure of €32.65 per month for outpatient doctors has been calculated.

Author Response

I have only a few minor comments:
Although the title will be longer, I would recommend to write out "NTM". The title would then be: Intersectoral cost of treating pulmonary non-tuberculous mycobacteria disease (NTM-PD) in Germany - a change of perspective in disease management

Reply: Done as requested.

General comment: Once the abbreviations of the drugs have been provided the authors should use these abbreviations in the text continuously and not jump from abbreviations to a drug's full name and vice versa  

Reply: Done as requested throughout the text.

Introduction, third para: One of the characteristics of pulmonary NTM disease is its high mortality. The authors do not mention any word on this fact and should therefore insert a sentence in order to complete the presentation of the high burden of NTM disease also in high income countries.

Reply: Thank you for that suggestion. We have now inserted a sentence addressing the high mortality.

Introduction, fourth para: Do the “unpublished data” refer to the dataset of reference no. 7? If so, please clarify!

Reply: Done as requested.

Basic assumptions for calculating drug costs of outpatients, first para: The authors should mention that possible discounts for the drugs of interest in favor of the statutory public health insurances are not included into the calculations.

Reply: Done as requested. We have now inserted the following sentence: “Discount agreements for drug costs, an option open to single health insurance organizations under the German system, are not considered in our calculations.”

Hospital management, second last para: Firstly, the authors should provide the full name of “OPS” not only the abbreviation. Secondly, the authors should give some examples of the mostly relevant OPS and provide the numbers of their associated numbers, e.g. for bronchoscopy, thoracic CT etc.

Reply: Done as requested.

Discussion, first para: The author should provide a short explanation how the figure of €32.65 per month for outpatient doctors has been calculated.

Reply: Thank you, we have now explicitly clarified the calculation of costs per month for outpatient doctors’ services and the calculation of the percentage share of the weighted total amount of outpatient costs in a new para in the Results section.

Round 2

Reviewer 2 Report

The little comments added to the paper improve the previous one and now it is more crealy what was the scope of this research and what results you obtained and how can we interpretate this results.